# The Biology of *Casmara subagronoma* (Lepidoptera: Oecophoridae), a Stem-Boring Moth of *Rhodomyrtus tomentosa* (Myrtaceae): Descriptions of the Previously Unknown Adult Female and Immature Stages, and Its Potential as a Biological Control Candidate

**DOI:** 10.3390/insects11100653

**Published:** 2020-09-23

**Authors:** Susan A. Wineriter-Wright, Melissa C. Smith, Mark A. Metz, Jeffrey R. Makinson, Bradley T. Brown, Matthew F. Purcell, Kane L. Barr, Paul D. Pratt

**Affiliations:** 1USDA-ARS Invasive Plant Research Laboratory, Fort Lauderdale, FL 33314, USA; susan_wineriter_wright@yahoo.com; 2USDA-ARS Systematic Entomology Lab, Beltsville, MD 20013-7012, USA; mark.metz@usda.gov; 3USDA-ARS Australian Biological Control Laboratory, CSIRO Health and Biosecurity, Dutton Park QLD 4102, Australia; jeff.makinson@csiro.au (J.R.M.); Bradley.brown@csiro.au (B.T.B.); matthew.purcell@csiro.au (M.F.P.); 4USDA-ARS Center for Medical, Agricultural and Veterinary Entomology, Gainesville, FL 32608, USA; kane.barr1@gmail.com; 5USDA-ARS, Western Regional Research Center, Invasive Species and Pollinator Health Research Unit, 800 Buchanan Street, Albany, CA 94710, USA; paul.pratt@usda.gov

**Keywords:** *Casmara subagronoma*, *Rhodomyrtus tomentosa*, stem borer, biological control of weeds, Myrtaceae, Gelechioidea

## Abstract

**Simple Summary:**

*Rhodomyrtus tomentosa* is a perennial woody shrub throughout Southeast Asia. Due to its prolific flower and fruit production, it was introduced into subtropical areas such as Florida and Hawai’i, where it is now naturalized and invasive. In an effort to find sustainable means to control *R. tomentosa,* a large-scale survey was mounted for biological control organisms. During these surveys, we found *Casmara subagronoma*, a stem-boring moth in Hong Kong and began to test its host affinity to *R. tomentosa*. *Casmara subagronoma* is only known from two male specimens from Vietnam and Indonesia, so in addition to host range tests, we also describe the female and immature life stages in this manuscript. *Casmara subagronoma* completes its life cycle in nearly 18 months and requires large, whole plants to do so. While *R. tomentosa* is likely the preferred host, *C. subagronoma* also completed its development on *Myrcianthes fragrans*, a Florida native, and *Myrtus communis*. The knowledge gained about this genus and its biology are quite valuable, but *C. subagronoma* will not be pursued for a biological control agent due to its long life cycle, difficult rearing protocols, and potentially broad host range.

**Abstract:**

*Rhodomyrtus tomentosa* is a perennial shrub native to Southeast Asia and is invasive in South Florida and Hawai’i, USA. During surveys of *R. tomentosa* in Hong Kong from 2013–2018 for potential biological control agents, we collected larvae of the stem borer, *Casmara subagronoma*. Larvae were shipped in stems to a USDA-ARS quarantine facility where they were reared and subjected to biology studies and preliminary host range examinations. *Casmara subagronoma* is the most recent *Casmara* species to be described from males collected in Vietnam and Indonesia. Because the original species description was based on only two male specimens, we also provide a detailed description of the female, egg, larva, and pupa. Finally, we conducted preliminary host range trials utilizing *Myrtus communis*, *Myrcianthes fragrans*, and *Camellia sinensis*. *Casmara subagronoma* emerged from *M. fragrans,* a Florida-native shrub, and larvae were able to survive in non-target stems for over a year (>400 days). Based on these findings and difficulty in rearing, we do not believe *C. subagronoma* is a suitable insect for biological control of *R. tomentosa* at this time, but may warrant further study. This investigation also illustrates the importance of host surveys for conservation and taxonomic purposes.

## 1. Introduction

*Rhodomyrtus tomentosa* (Aiton) Hassk. (Myrtaceae) is an evergreen shrub native throughout Southeast Asia and invasive in Florida and Hawai’i [1,2]. *Rhodomyrtus tomentosa* (downy rose myrtle) was widely exported from Southeast Asia, beginning in the late 1800s primarily for use as an ornamental plant, but also for its edible fruits [3] (Figure 1). More recently, *R. tomentosa* has been the target of extensive speculation for medicinal extracts [4].

Downy rose myrtle escaped cultivation in Hawaii and Florida by at least the 1940s, and by the 1960s was a proposed target of biological control efforts [5,6]. Extensive surveys have been done throughout the native range in China, Hong Kong and Thailand [7,8,9]. Several new insects have been described and several of these have been investigated for their suitability in biological control [7,8,9,10,11,12,13]. However, each of these has proven to be insufficiently host-specific and therefore unsuitable for use in the biological control of downy rose myrtle.

*Casmara subagronoma* Lvovsky (Lepidoptera: Gelechioidea: Oecophoridae) is prevalent on mature *R. tomentosa* throughout Hong Kong including Lantau Island, Shing Mun and Sai Kung (J. Makinson, personal communication). At *R. tomentosa* survey sites, other shrub species showing similar damage patterns were often surveyed, but *C. subagronoma* has not been recorded from other species. All described *Casmara* larvae feed on the pith and xylem, leaving in their wake dead hollowed stems (Chen, 1958 and others). Hollowed branches break easily, revealing the bore holes. At periodic short intervals, larvae feed deeper into the sides of stems, but not to the exterior, forming round sinks for the deposit of frass and are subsequently closed off by webbing. Live woody stems provide food and shelter to developing larvae. Most species take many months or years to complete their life cycle.

Twenty-one stem borer species have been described and placed in the genus *Casmara* Walker 1863 (Lepidoptera: Gelechioidea: Oecophoridae). Species in the genus range through eastern Asia (China, Japan, Korea) [14,15,16,17,18,19,20,21,22], Southeastern Asia (India, Indonesia, Malaysia, Thailand, and Vietnam) [22,23,24,25,26,27,28], New Guinea [29], and Northern Queensland, Australia [30]. The last species identified and described was *Casmara subagronoma* Lvovsky based on two males, one from Sumatra, Indonesia (collected in 1982) and another from Northern Vietnam [22].

Host associations for only three species of *Casmara* are reported. *Murraya exotica* L. (Sapindales: Rutaceae) is a host of *Casmara kalshoveni* Diakonoff [27,31]. *Camellia oleifera* Abel and *Camellia sinensis* (L.) Kuntze (Theales: Theaceae) are hosts of *Casmara patrona* Meyrick [14,17,19,32] and *Casmara agronoma* Meyrick [19,33]. *Casmara patrona* and *C. agronoma* cause economic damage to both *Camellia oleifera* [32,34] and *Camellia sinensis* [17,19,33], economically important crops (tea) in China and India [35].

Chen [17] reported the following on *C. patrona* regarding tea shrubs (*Camellia* sp. not specified). *Casmara patrona* had one generation per year, adults emerged in May, June, and neonate larvae fed into the tips of plants, and downward into the pith and xylem of green shoots, woody twigs, and branches (hereafter called stems). Larvae, when disturbed, moved rapidly back and forth in the smooth-walled, hollowed-out stems. At periodic short intervals, larvae fed deeper into the sides of stems, but not to the exterior, forming round sinks. Smaller stems broke easily, revealing bore-holes, a natural occurrence during windy days. Larvae overwintered in hollowed-out stems. Pupation, which occurred in the hollowed-out stems, was further protected by flocculent plugs of larval webbing. In young shrubs, larval boring killed adjacent unattacked stems. In mature shrubs, larvae bored stems near to ground level. Li et al. [32] conducted field and laboratory studies of *C. patrona* in *Camellia oleifera*. Field experiments allowed them to cage active stem borer stems on whole plants by collecting and removing frass daily. When frass production ceased, stems were cut and taken to the lab where they were monitored for adult emergence.

Li et al. [32] found that *C. patrona* goes through five larval instars. Most *C. patrona* larvae overwintered once, and the remaining larvae twice. Larvae overwintered as third–fifth instars. Those larvae that overwintered once had a low percent of parasitism and those that overwintered twice had a higher instance of parasitism. Development times for *C. patrona* ranged from 285–310 days (mean = 300 ± 2.61 days). *Casmara patrona* larvae cause wilting of leaves and shoots of *Camellia oleifera* after larvae bored past 4–6 leaves and death of shoots after about 40 cm of stem-boring [36]. Xiong [34] reported 95% of *C. patrona* damage could be controlled by clipping tea tips in early–mid August.

Determining the fundamental host range of biological control organisms involves exposing potential agents to non-target species starting with the closest relatives and moving outward systematically [37,38,39]. This method ensures relatively little risk that, once released, an agent will not cause significant harm to native plants or economically important species within its host’s invaded range [39,40]. In the case of *C. subagronoma*, this would include native members of the Myrtaceae. However, because *C. subagronoma* is potentially closely related to pests on *C. sinensis* in its native range, and *C. sinensis* is a non-invasive ornamental in the Southeastern USA and more recently a commercially grown crop in South Carolina, USA [41,42], this species was also presented as an initial screening host to determine further pursuit of *C. subagronoma* as a biological control agent. While this approach does not explicitly follow Briese (2003) [39], it is consistent with other testing methodology. For example, the Technical Advisory Group for the Biological Control of Weeds (USDA Animal Plant Health Inspection Service) now advocates in their manual for testing hosts of related insect taxa if those hosts are imperiled or economically important [43].

Herein, we describe a potential biological control agent, *C. subagronoma*, including the first descriptions of the adult female and immature life stages. We also roughly describe larval development in *R. tomentosa* stems, based on indirect observations, and provide results from initial host range tests.

## 2. Materials and Methods

### 2.1. Collection

Damaged *Rhodomyrtus tomentosa* stems, usually woody twigs and branches ca. 3–5 mm × 15–30 cm (diameter × length), each stem potentially containing one *Casmara subagronoma* larva, were collected from various locations in Hong Kong (Figure 2) and shipped under USDA APHIS PPQ permit nos. P-526P-09-02486, 13-00144, 13-00151, 15-03820, and 15-04633 to the USDA Agricultural Research Service Invasive Plant Research Laboratory in Gainesville, Florida USA (Florida Biological Control Laboratory, FBCL hereafter) from March 2013 through March 2018. Identity of larvae in stem collections in 2013 were unknown when received. Adults reared were identified to genus by J. E. Hayden, Curator of Lepidoptera, Florida State Collection of Arthropods, Florida Department of Agriculture and Consumer Services, Division of Plant Industry (FDACS, DPI), Gainesville, FL, USA. M.A. Metz (coauthor) identified the species. *Casmara subagronoma* specimens described herein were preserved from 2014–2017 field-collected larvae and from field-collected larvae lab-reared to other stages.

*Rhodomyrtus tomentosa* plants were obtained from seeds, field-harvested cut stems, and whole plants. Seeds were collected from plants in Cypress Creek Preserve, Palm Beach County, Florida. They were started in Pro-Mix FPX and transferred to 11-L pots with Pro-Mix BX potting mix (PT Horticulture, NC, USA). Cut stems and whole plants were harvested from Lake Lizzie Nature Preserve, Oseola Co, FL, and in Yucca Pens Unit Wildlife Management Area, Cape Coral, FL. Collected whole plants were potted similarly to plants grown from seeds. *Myrcianthes fragrans* and *Camellia sinensis* plants were obtained from San Felasco Nursery, Alachua, FL and *Myrtus communis* plants and cut stems were obtained from O’Tooles Herb Farm in Madison, FL and the USDA-ARS Western Regional Research Center, Albany, CA, USA. All whole non-targert plants were potted in 11-L pots with Pro-Mix BX potting mix and housed in shaded screen houses in Gainesville, FL, USA with 30% shade cloth. All plants were watered daily and fertilized every six months with slow release fertilizer (Osmocote 19-5-8, A.M. Leonard, OH, USA). During the natural dormancy period of plants in North Florida, two Beamflicker™ lights, each with a 600-w high pressure sodium bulb (HPS), were utilized in the larger screened house to keep them actively growing. The screened house was winterized from December to March by covering it with two, air-separated layers of plastic and regulating the temperature with a thermostat-controlled gas heater, vent, and wall-mounted shuttered exhaust fan.

### 2.2. Sample Preparation for Descriptions

Additional information about specimens can be found in the online database: USNM Department of Entomology Collections (http://collections.nmnh.si.edu/search/ento/; indexed by the unique identifier USNMENTXXXXXX). M.A.M. dissected and prepared genitalia from pinned specimens following the methods of Clarke [44] and Robinson [45]; took measurements with an ocular micrometer from the left side of the specimen when possible; and used a Visionary Digital imaging station (Dun, Inc., Palmyra, VA, USA) for photographs and the Gnu Image Manipulation Program (GIMP.org) for photo-editing. Morphological terminology follows Hodges [46,47], Kristensen [48], and Patočka and Turčáni [49].

### 2.3. Rearing

For preliminary tests, 16 larvae from a 2015 field collection (2015-1008), 12 larvae from a 2016 collection (2016-1001), and 18 larvae from two collections in 2018 (2018-1001,1002) were utilized. Additional larvae from these collections were reared for possible establishment of a colony.

Field-collected larvae were shipped within the stems in which they were found feeding. A number was assigned to each stem, upon arrival in the order in which it was removed from the shipment container (FBCL-YEAR-XXXX, hereafter YEAR-XXXX). Most stems contained one larva/stem; a few stems were empty and these account for omitted numbers in the tables. Some early shipments of larvae were maintained in their Hong Kong stems until they had nearly or completely bored through them due to limited knowledge of how to care for larvae. Later shipments, including those for preliminary tests were transferred to cut stems of *R. tomentosa* or, to stems on whole plants of either *R. tomentosa*, or *C. sinensis* upon receipt. Transfers upon receipt became necessary to ensure that larvae had survived conditions of international transit and could be expected to feed freely throughout their development time. Additional larvae from earlier and later collections were reared for possible establishment of a colony.

To transfer, each larva was extracted from its Hong Kong stem (Figure 3) while viewing through a dissection microscope and carefully paring away the cork on one side of the stem with an Exacto knife. When the bored tunnel was exposed enough, the larva was either lifted out with a small paintbrush or coaxed to move out of the tunnel onto a flat surface with gentle touches of the brush (Figure 3). The larval head was then placed into the distal end of a fresh stem with a manually drilled hole slightly larger in diameter than the bore in the proximal end of the larva’s previous stem and the length of its body. Drill bits ranging in size from 1.5–4.8 mm were used to measure the proximal bore size of the old stem and to make a distal bore in the new stem.

Individual stems were placed in cuboid, clear rearing containers with slip-fit lids (www.pioneerplastics.com) containing a capillary mat cut to fit flat on the bottom. In addition, there was included a petri dish, 60 × 15 mm (diameter × height), placed under the distal end of the stem bore to collect the frass ejected, and an inverted 30-mL cup (WNA Comet P -10, 1-oz portion cup) filled with tap water and inverted on the petri dish lid fitted with two organic cotton rounds (Swisspers, www.uscotton.com), to elevate humidity. Rearing containers were not air-tight and the combination of the capillary mat, collected wood-like frass, and continuous water-soaked cotton worked well to keep stems hydrated and the conditions within the containers at about 90% RH. Rearing containers were held in a Percival Environmental Chamber programmed as follows: 16 L (with a 20 min ramped sunrise and sunset); 27 °C and RH at 70% during the light period, 22 °C and 65% during the dark period. A few 2015–2016 long-lived larvae, more than a year after initiation of their tests, were transferred to whole plant stems and held under full spectrum lights in the laboratory at 25–27 °C or in the environmental chamber under the above conditions.

Whole plants with 2018 larvae were held in a greenhouse at 27 °C with supplemental light provided by full spectrum fluorescent bulbs to provide 16L. Additional light was provided by a Beamflicker™ 400-w high-pressure sodium bulb (HPS) (Parsource, Petaluma, CA, USA). Relative humidity was maintained at approximately 55% by a hanging Jaybird Aquafog direct feed fogger (www.jaybird-mfg.com, series 700). Stems with pupae were removed from potted plants and held in the environmental chamber until adults emerged (Figure 4). If no male emerged synchronously with a female, her eggs (Figure 3) were collected and counted when oviposition occurred in an individual rearing container (see Appendix A for additional information on *C. subagronoma* lab life cycle on *R. tomentosa*).

### 2.4. Preliminary Host Range Examination

Host range tests are generally conducted with laboratory colonies of insects that have been cleared of pathogens and parasites. However, because of the difficulty rearing these stem-boring insects and their relatively long life cycles, we elected to test field-collected larvae from two shipments, 2015-1008 and 2016-1001, on cut stems of *M. fragrans* and *M. communis* (Tests I and II) and from two shipments on whole plants of *C. sinensis* (Test III).

Tests I and II: Larvae for Test I were collected from Ngong Ping, Lantau Island, Hong Kong, 23–27 November 2015 (late autumn), received in Florida on 1 December 2015, and designated shipment 2015-1008. Larvae for Test II were collected from Luk Wu, Sai Kung Peninsula, 5–7 March 2016 (early spring), received in Florida on 9 March 2016, and designated shipment 2016-1001. For Test I, 2015-1008 larvae, ten of 16 larvae were reared 49 days on *R. tomentosa* cut stems before test initiation on 19 January 2016 of *M. communis* cut stems; and six of 16 larvae were reared for 75 days on *R. tomentosa* cut stems before test initiation on 16–17 February 2016 of *M. fragrans* cut stems. For Test II, 2016-1001 larvae, 12 of 12 larvae were reared for 189 days on *R. tomentosa* cut stems before test initiation on 14 September 2016 of both *M. communis* and *M. fragrans* cut stems. Tests I and II larvae were reared on Florida *R. tomentosa* cut stems first, to acclimatize them to laboratory environmental chamber summer conditions and to obtain the same size larval groups before testing. The variability in test design reflects the seasonal and size variations of field larvae. Larvae for both tests were assigned to tests using a random number generator. Test larvae (*N* = 4 from Test I and *N* = 6 from Test II), surviving on cut stems for 375 days or more were transferred to whole plants on 26–28 September 2017 for the remainder of their test life. Plants were held in the environmental chamber and/or laboratory, conditions described above.

Test III: Stems for Test III were collected on 12 March 2018. Most stems were received on 15 March 2018 (2018-1001) and a few stems 19 March 2018 (2018-1002), both herein referred to as 2018-1001,02 due to the same date of collection. The smallest stems with potential larvae were reserved for Test III. These stems were assigned a number prior to opening them. Several stems were empty upon receipt, had a different stem borer, *Zeuzera* sp. (Lepidoptera: Cossidae), or were otherwise non-viable. These reasons account for the apparent irregularities in the larval numbering presented in the results. In all, 18 larvae were available for Test III. Seventeen larvae were collected from Ngong Ping, Lantau Island, and one larva was collected from Luk Wu, Sai Kung Penisula. Six additional non-test larvae from larger stems (and listed in the six last columns of Appendix A) were transferred to whole plants of *R. tomentosa* for possible colonization: Two of these larvae were collected from Ngong Ping, near Shing Mun, Central New Territories; two larvae from Lung Mun, Sai Kung Peninsula; and two larvae from Luk Wu, Sai Kung Peninsula.

Test III larvae were transferred to stems of whole plants, one larva/stem of *R. tomentosa* or *C. sinensis* (*N* = 9 larvae/species). Larvae were distributed among stems of three plants of *R. tomentosa* and four plants of *C. sinensis*. As larvae were removed from their field stems, they were alternately transferred to whole plants. Six larvae were transferred to whole plants on 15 March (*N* = 3 larvae/species), ten larvae on 16 March (*N* = 5 larvae/species), one larva to *R. tomentosa* on 23 March, and one larva to *C. sinensis* on 19 March. Each larva-containing stem was caged with a small net cage to catch frass and each plant was then placed into a large net cage (BugDorm 2400F, BugDorm Store, Taiwan, China). For some stems, side shoots and leaves had to be removed to position the small net cages. Following removal of a larva, the field stem was placed in a plastic bag to maintain humidity. When all larvae had been transferred to whole plant stems, the outside diameter of the proximal end of each field stem and its respective bore diameter were measured using a Rok 0–150 mm digital caliper (Roktools, Guangdong, PRC). Larvae bore tight-fitting tunnels, and the diameter of bores was used as an indirect quantification of larval size, i.e., pronotal width.

### 2.5. Tracking Larval Size and Development

Larval frass was collected from the small net cages encasing the top portion of each stem beginning 13 April 2018, ca. 4 weeks from the collection date, and continuing every two weeks thereafter with the last collection made on 14 September 2018 or at ca. 26 weeks. Frass was dried at 60 °C for at least 72 h and weighed to provide patterns of feeding and quiescence. Visual observations of frass pellet size every two weeks were compared to frass collections of other larvae that had been reared and whose lifetime total accumulation of frass had been sifted to discrete sizes using USA Standard Sieves, nos. 12–35 (largest frass pellet size-smallest frass size). These observations, though not recorded, provided an additional clue to where the larvae were in development. If little frass or no frass was found at six and again at eight weeks, the larva was presumed dead, the stem was cut, and the state of the individual recorded.

Ten weeks after test initiation (25–26 May, 2018) and at the two-week frass collection interval, the observation was made that some larvae had possibly bored or consumed the length of their stems and needed to be transferred. After frass was collected, stems with larvae were cut, opened and searched for larvae, and larvae transferred to stems of fresh plants. Proximal bore diameter, length of bored tunnels, length of bored stems, and length of any dead leaves and shoots on stems that had resulted from boring were all measured. If pupae were found within stems, each of these stems was placed in a piece of 1.7-mm diameter black polyvinyl chloride irrigation tubing, with the end opposite where an adult might emerge plugged with cotton. Each plastic tube-encased stem was placed in a rearing container similar to larval rearing containers. Pupal containers were placed in the environmental chamber (settings above), and watched daily for adult emergence. Date of adult emergence and sex were recorded. On 18 September 2018, all plants and stems were moved to the USDA-ARS Invasive Plant Research Laboratory quarantine greenhouse in Fort Lauderdale, FL, USA, where they were held until 22 August 2019. All stems were then harvested and dissected to inspect for larvae or pupae.

## 3. Results

### 3.1. Species Description

#### 3.1.1. Description of Adult Female

*Casmara subagronoma* Lvovsky, 2013 (urn:lsid:zoobank.org:act:9B90DC7C-C38F-4604-B20B-F6E97D1D3B06 (Figure 5, Figure 6, Figure 7, Figure 8, Figure 9 and Figure 10).

Similar to male, slightly larger on average based on few specimens. Genitalia papillae anales very membranous with long setae, barely demarcated compared to surrounding terminal membrane; apophyses anteriores straight, slightly capitate, apex reaching posterior margin of VII (Figure 5A,B); apophyses posteriores length 2× length of anterior apophyses, straight, apex slightly tapered; ventromedial surface of segment VIII around ostium undifferentiated; ostium width slightly less than 1/3 width of anterior margin of segment VIII, antrum length 2× width, cylindrical at entrance then expanding slightly with right and left, internal denticulate, flattened discs, tapering anteriorly; ductus bursae as wide as antrum then taper to entrance of corpus bursae, short, length 2× width; junction of ductus bursae and corpus bursae indistinct; corpus bursae oblong, widest diameter 1/3 length of ductus bursae; signum on the ventral wall of corpus bursae, a longitudinal strip of denticles, outer margin slightly irregular, with an indistinct, smoother channel down the center.

#### 3.1.2. Larval Diagnosis

Last instar cylindrical (Figure 6A), blunt anteriorly, and tapered posteriorly in dorsal view; integument wrinkled, milky white, surface lacking coloration except hardened cuticle of head capsule, thoracic shield, spiracles, pinacula, anal shield, and proleg base, light to dark brown; prothoracic shield and anal shield well-developed; T1, A7, and A8 spiracles much larger, vertical; all primary setae arising from weak pinacula. Head (Figure 6B): Hypognathous, mostly light brown, dark brown anteriorly and ventrally, lacking any noticeable pattern, surface with sculpturing composed of raised ridges defining juxtaposed, enclosed polygonal shapes; epicranial suture length approximately 10× greater than height of frontoclypeus; frontoclypeus height only slightly greater than width at base; ecdysial line evident; six stemmata, 3–6 ventrad S1 and S2, 1–4 close to each other, 5 and 6 somewhat separated and ventrad, only 5 ventrad antenna; primary setae all represented, positioned more mediad and ventrad, P2 mediad P1 and ventrad vertical midline of head; mandible with four teeth, two dorsal teeth mostly fused, but discernible, one mandibular seta with socket on ventral surface set within a longitudinal channel on ventral surface of mandible. Thorax (Figure 6C): Prespiracular sclerite large and separate from prothoracic shield; prothoracic L group trisetose, setae arranged in a triangle with L1 and L2 anterad; T1 with two SV setae, T2 and T3 with one SV seta; T2 with L1 more dorsad L2. Abdomen (Figure 6E): A8 with L setae vertical, L1 and L2 on separate, but approximate pinacula, distance to L3 much greater, with 2 SV setae; A9 with L1 and L2 horizontal, on the same pinacula, L3 ventrad and further away, with 2 SV setae; A10 with D2 inserted below anal shield and pointing ventrally. Prolegs with crochets triordinal, circle incomplete and tapered medially, crochets of A10 transverse and complete (Figure 6D).

#### 3.1.3. Pupa

Head, antennal base, and prothorax patterned with a mosaic of crenulae, rest of body smooth or with slightly roughened texture; two pairs of setae anteriorly on head; labrum shield-shaped, lacking setae; labial palpus long, spindle-shaped; mandibula 1/3 length of oculus, trapezoid-shaped, slightly curved; maxillary palpus rectangular, posteriad eye margin, same width as oculus; haustellum visible to antennal convergence, to approximately distal third of A2; forefemur not visible; foreleg 1/2 length of forewing, extended just distad anterior margin of A2; midleg extended to anterior margin of A2, antennal sheath extended beyond posterior margin of A4; hindtarsus visible, not extended beyond posterior margin of A4; forewing extended to posterior margin of A4; vestigial prolegs barely evident; genital suture on A8 evident as a cephalocaudal slit anterior to lateral protrusions of A8: anal suture a cephalocaudal slit, posteroventral. Thoracic spiracle simple without spines or specialized collar; abdominal spiracles simple; A9–Al0 fused, no suture line visible; Tl-A8 with dorsal and lateral setae; A9 with a dorsolateral seta; cremaster absent; terminal segment without terminal setae (Figure 7).

#### 3.1.4. Egg

Length 1050 μm. Dark reddish-brown, length 1.4× greater than widest point, bilaterally symmetrical, heart-shaped, width of end opposite micropile 2× width of micropile end (Figure 8A). Chorion with a longitudinal groove bisecting wide end into right and left hemispheres, surface sculpturing composed of blunt points connected radially and longitudinally forming radial and longitudinal rows of shallow pits, height of points greatest around micropile and diminishing longitudinally towards wider end of egg, raised edge heights reduced, leaving only a roughened texture adjacent to longitudinal groove (Figure 8B,C). Micropile flat depression surrounded by a circular arrangement of blunt points (Figure 8D).

#### 3.1.5. Specimens Examined

China: Hong Kong: Adults: Collected as larva feeding in woody stem of *Rhodomyrtus tomentosa*: 6 ♀♀ (USNMENT01328209, USNMENT01328218, USNMENT01328221, (USNMENT01328222, USNM slide # 146,479), USNMENT01328223-24) Ngong Ping: MF Purcell: 26.XI.2014; 1 ♂ (USNMENT01328219, USNM slide # 146,476) Lantau Island: Ngong Ping or other location: 7-8,12.I.2015: BT Brown; 1 ♀ (USNMENT01328215) Lamma: 11.I.2015: BT Brown; 1 ♀ (USNMENT01328207) Lantau Island or Sham Tseng: 27.III.2014: J Makinson; 1 ♀ (USNMENT01328205, USNM slide # 146,478) Wu Kau Tang: 18.V.2015: BT Brown; 1 ♀ (USNMENT01328216, USNM slide # 146,477), 1 ♂ (USNMENT01328210, USNM slide # 146,475) Lantau: 12.I.2015: BT Brown. Larvae: 4 specimens (Lot # USNMENT01328213) Lantau: 29.II-5.III.2016: BT Brown. Pupa: 1 female (USNMENT01200873) Lantau Island (FBCL 2017 1002 #1, Received Q 27.11.2017, Collect 28.IV.17 KBARR).

Additional material also came from China: Hong Kong: Larvae: 1 specimen (USNMENT01328220) Lantau Island: Ngong Ping: 12.VI.2015: MF Purcell; 4 specimens (Lot # USNMENT01328208), 5 specimens (Lot # USNMENT01328214), 6 specimens (Lot # USNMENT01328211), 7 specimens (Lot # USNMENT01328217) Lantau Island/Sham Tseng: 27.III.2014: J Makinson; 10 specimens (Lot # USNMENT01328233) Lantau Island/Sham Tseng: 24-27.XI.2015: BT Brown; 1 specimen (USNMENT01328212), 3 specimens (Lot # USNMENT01328206) Sai Kung: 29.II-5.III.2016: BT Brown (Figure 9).

Lvovsky (2013) diagnosed *C. subagronoma* based on two males; the holotype from Vinh Phuc Province, Vietnam and a single paratype Indonesia. He noted:

“[*Casmara subagronoma*] is similar to *Casmara agronoma* Meyrick, 1931, described from China, but can be distinguished by a large pale gray spot near the outer margin of forewing (this spot is absent in brown wing of *C. agronoma*). Male genitalia of the new species differ in a narrow apical part of gnathos (1/3 the length of the rest part of gnathos) and a long notch at the distal half of aedeagus which is approximately equal to half of the aedeagus length. Whereas in *C. agronoma*, narrow apical part of gnathos is 1/2 the length of the rest part of gnathos, and the notch in the distal part of aedeagus shorter than half of the aedeagus length.”

We corroborate the diagnostic features of the wing pattern (Figure 9) and male genitalia (Figure 10A,B). We found in the female genitalia that the two species are remarkably similar, but the signum of *C. subagronoma* is half the length of the signum of *C. agronoma*.

### 3.2. Casmara Subagronoma Development and Initial Host Range Test

In Tests I and II, several larvae survived for well over a year on *M. fragrans* and *M. communis*. One male (larval ID 2015-1008-04) emerged from *M. fragrans* 160 days after transfer and a second male (larval ID 2016-1001-15) emerged 267 days after transfer (Table 1). A larva (larval ID 2015-1008-43), alive for more than 422 days after transfer, reached pupation in *M. fragrans*, but was found to have oriented itself in the wrong direction, died, and was infested with mites at 475 days (Table 1). No adults emerged from *M. communis*, but 4 of 8 survived as larvae for >200 days. In Test I with 2015-1008 larvae that tested larvae on *M. communis* and *M. fragrans*, only one larva completed development on *R. tomentosa* (larval ID 2015-1008-31). In Tests I and II, a total of ten larvae were alive (*R. tomentosa* N = 7, *M. communis N* = 2, *M. fragrans N* = 1) and feeding for >1 year and were eventually transferred to whole plants in September 2017 (laval IDs 2015-1008-9,16,31,38 and 2016-1001-2, 6,8,10,18,22).

In Test III, nine larvae (100%) on *R. tomentosa* were alive six weeks after transfer while all nine larvae on *C. sinensis* died within the same time (Table 2). Larvae transferred to *C. sinensis* showed no measurable boring or feeding activity past our manually drilled holes, whereas all larvae transferred to *R. tomentosa* showed signs of significant excavation beyond the initial drill point (Table 2). Two of nine (22%) test larvae emerged as adults from *R. tomentosa* stems (2018-1001,02-15,20). Of six larvae reared like test larvae for potential colonization, four of six larvae (66.6%, largest larvae in 2018–1001,02 shipments according to field stem bore diameter) emerged as adults (2018–1001,02-1,3,5,6,33,34); one of six larvae (16.7%, 2018-1001,02-3) was parasitized by *Macrocentris sp.,* and one of six larvae died after 25 May 2018 possibly due to transfer injury or while molting.

Stem diameter had a positive, though weak and non-significant relationship to bore diameter; larger larvae were generally in larger stems (R^2^ = 0.23, *p* = 0.19). Larval frass production by weight progressed in steps throughout the development (Figure 11). Generally speaking, with each molt, the larva produced more frass, but each molt was punctuated by a steep decline in frass production, eventually culminating in a termination of all production (and consumption) when the larva pupated (or died). Death and pupation could sometimes be differentiated by the size of the frass preceding the cessation in production. If frass size was relatively large (e.g., could not be sifted through a #12 sieve), the larva had likely reached pupation.

In successive weeks, six of nine (0.67%) larvae feeding on *R. tomentosa* survived 10–15 weeks. Three of six emerged as adults (0.33%) in June or July of 2018; one of six (17%) was accidentally injured during excavation for stem transfer and died between 10–12 weeks after the initiation of the trial. Five of six continued to live as larvae a minimum of 22 weeks, but bored to the bottom of a root ball and died (Table 1). In all shipments, a few larvae were parasitized and *Macrocentrus* spp. (Hymenoptera: Braconidae: Macrocentrinae not *M. linearis*, R. R. Kula, USDA-ARS Systematic Entomology Laboratory) emerged from several of these.

## 4. Discussion

In the course of conducting extensive surveys for potential biological control agents, many arthropods were discovered, some of which may be new to science or have scant information associated with them. Invertebrate diversity was often overlooked when applying conservation policies, simply because many of the taxa are either undescribed, unknown, or understudied [50]. Our efforts to find suitable agents for controlling *R. tomentosa* shed light on how several insects utilize this resource on Mainland China and Hong Kong SAR, information that was previously unknown and can now be utilized to justify conservation efforts or further research on the host and consumer associations. In this specific case, all that was known about *C. subagronoma* was from two male adult specimens. The addition of rearing data, occurrence data, and descriptions of the female, larval, and pupal characteristics drastically increased our understanding of this species and the genus.

Initial host range observations indicated that *C. subagronoma* may have some obligate host association with members of the Myrtaceae, but this level of specificity was far too broad to be acceptable for use in biological control. We observed boring and stem consumption in mid-instar larvae transferred to *M. fragrans* and *M. communis* and emergence from *M. fragrans.* Conversely, we observed few or no signs of feeding past our manually drilled stem hole by early instar larvae that were transferred *C. sinensis*. We can easily rule out overwintering or metamorphosis in the larvae transferred to *C. sinensis* because they were not mature enough for this to occur. All larvae on *R. tomentosa* produced frass for weeks beyond their transfer to a large potted plant in the Florida quarantine facility. Though this was by no means a comprehensive host range test, it suggests that *C. subagronoma* has a decidedly different host range than *C. agronoma* or *C. patrona* that feed on *Camellia* species and that it is likely a specialist on myrtaceous flora. Though larvae that were hand-transferred to *M. fragrans* were able to complete development, we were unable to test the oviposition preference between *R. tomentosa* and *M. fragrans* or other species. Female ovipositional preference may have further constricted the host range because larvae would not engage in host-finding behavior [51]. Though we were unable to test ovipositional behavior, our examinations show that *C. subagronoma* behaved quite similarly on *R. tomentosa* to *C. patrona* on *Camellia* plants. This could have implications if *R. tomentosa* is utilized for any type of commercial venture.

Though its ecological host specificity is unknown and potentially a barrier, the lifecycle of *C. subagronoma* further presented challenges for its use as a biological control agent. Larvae required the structure and moisture provided by live stems to progress through development; therefore, transferring them between stems was quite difficult and often fatal for the larva. Additionally, many larvae that successfully emerged did so after 16–22 months of development. This step would itself potentially be an insurmountable challenge to pursuing biological control—host range testing would take at least this long and multiple generation testing would further extend the length of time needed. Lepidopteran stem borers are frequently targets of biological control efforts due to their propensity to be pests of cereals (rice, wheat, barley, sugarcane) and other crops (coffee, tea) [52], but are not common among biological control agents. Other borers including root miners have been used extensively in biological control efforts. These include *Rhinoncomimus latipes* (Coleoptera: Curculionidae) which attacks mile-a-minute vine (*Persicaria perfoliata* (L.) H. Gross (Polygonales: Polygonaceae), *Neochetina bruchi* Hustache and *N. eichhorniae* (Coleoptera: Curculionidae) that bore into water hyacinth stems (Liliidae: Pontederiaceae: *Eichhornia crassipes* (Mart.) Solms), *Agapeta zoegana* (Lepidoptera: Tortricidae) bores into various *Centaurea* species (Asterales: Asteraceae) roots, and the alligator weed stem-borer moth, *Arcola malloi* (Pastrana, 1961) (Lepidoptera: Pyralidae) causes the stems of alligator weed (Caryophyllales: Amaranthaceae: *Alternanthera philoxeroides* (Mart.) Griseb.) to collapse and sink [53]. In terms of woody plants, *Agonopterix assimilella* (Lepidoptera: Oecophoridae) begins as a stem-boring caterpillar and then moves outside to feed on green stems of *Cytusis scoparius* (L.) Link (Fabaceae) (scotch broom). *Agonopterix assimilella* (Treitschke, 1832) was released in 2007 and 2009 in New Zealand, but continues to be rare with little impact despite multiple introductions [54,55]. Perhaps the most successful example of a lepidopteran stem borer utilized as a biological control agent is *Neurostrota gunniella* (Busck) (Lepidoptera: Gracillaridae), released to control *Mimosa pigra* L. (Fabaceae) in Australia [56]. Damage inflicted by internal larval feeding drastically reduced leaf cover of the spiny shrub (Smith and Wilson 1995). The fundamental difference between these examples and *C. subagronoma* was the drastic difference in life cycles (and perhaps host affinity). All of the above examples given are at least univoltine if not multivoltine with several generations per year (e.g., *N. gunniella, Neochetina* spp., *R. latipes*). Elucidating the biology of this insect further emphasizes the need to prioritize both weed targets and herbivores discovered during foreign exploration. Unfortunately with *R. tomentosa,* host-specific agents have not yet been found and *C. subagronoma* appears incompatible based on both host affinity and its biology.

## 5. Conclusions

One of the collateral benefits of biological control agent surveys is that new species and under-described species are often discovered in the process. While some of these species will prove safe enough for implementation into a biological control strategy for invasive species, many will not. *Casmara subagronoma* is insufficiently host-specific to be used for weed biological control, and requires significant efforts to cultivate in laboratory settings. However, elucidating its biology is important for understanding the biology of genus and providing support for conservation of *R. tomentosa*-dominated shrublands in Hong Kong.

## Figures and Tables

**Figure 1 insects-11-00653-f001:**
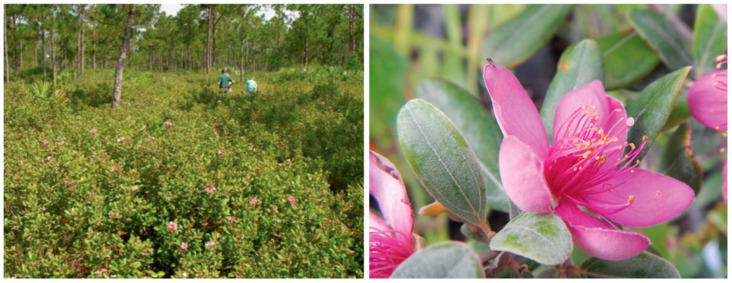
*Rhodomyrtus tomentosa* infestation in Palm Beach County, Florida, USA (left) and a close-up of the attractive flower from which is derives the common name, downy rose myrtle (right) (photos: M.C. Smith).

**Figure 2 insects-11-00653-f002:**
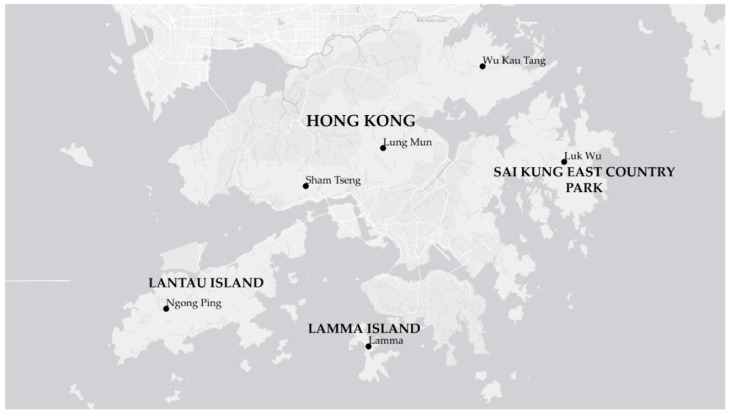
Collections were made at several points near these locations in Hong Kong between 2013 and 2018 as part of an effort to find suitable biological control candidates for *R. tomentosa*.

**Figure 3 insects-11-00653-f003:**
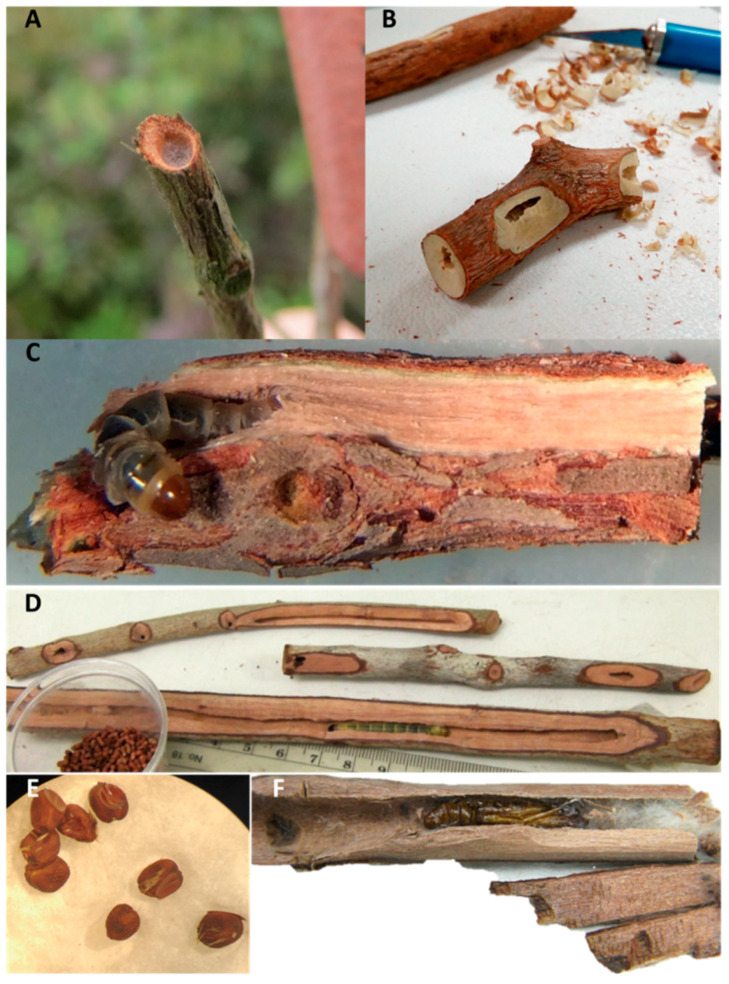
(**A**) Webbed *Casmara subagronoma* larval bore observed in the field in Hong Kong (photo J.R. Makinson). (**B**) Larval dissection of *R. tomentosa* stem showing bored tunnel ends, the consumed phloem and xylem of the stem, and a trapped larva ready for transfer (photo by E. Peralta). (**C**) Field-collected larva ready for transfer (photo E. Peralta, enhanced by E.J. Talimas). (**D**) Field-collected larva that reached late instar after 3 transfers. A 30-mL cup shows its collected frass (photo M.E. Clark, enhanced by E.J. Talimas). (**E**) Eggs (photo K.L. Barr). (**F**) Stem opened to reveal pupa and larval plug of webbing and frass used to plug hole and presumably to protect maturing pupa until adult emergence (photo S.A. Wineriter–Wright).

**Figure 4 insects-11-00653-f004:**
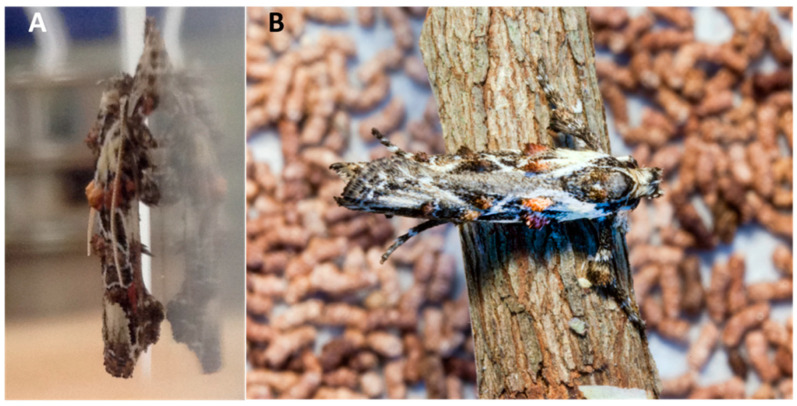
(**A**) Lateral view of a newly emerged *Casmara subagronoma* adult (photo K.L. Barr). (**B**) Dorsal view of a newly emerged adult resting on the stem, its last larval instar bored and from which the adult exited (larval ID 2014-1002-96). Frass is visible in the background (photo J.W. Lotz).

**Figure 5 insects-11-00653-f005:**
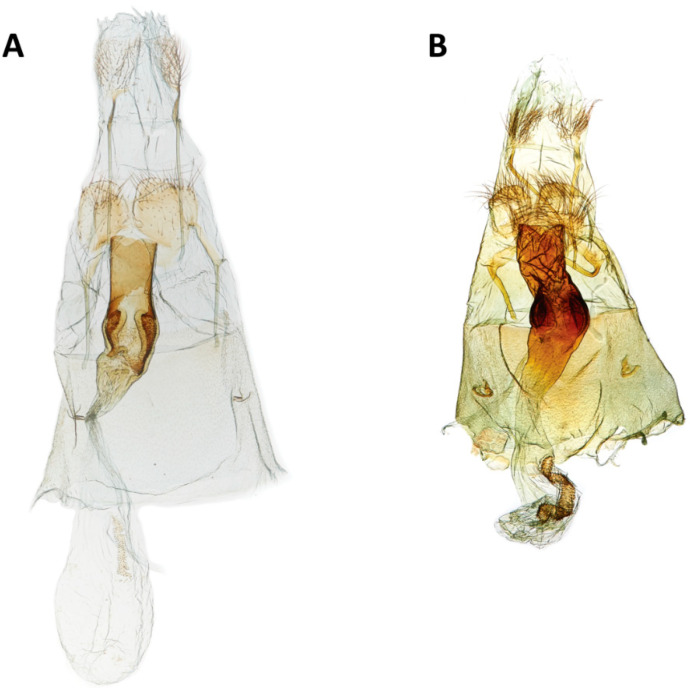
Female genitalia, ventral view. (**A**) *Casmara subagronoma* (USNMENT01328216, USNM slide # 146,477). (**B**) *C. agronoma* (USNMENT01200871, USNM slide # 146,473).

**Figure 6 insects-11-00653-f006:**
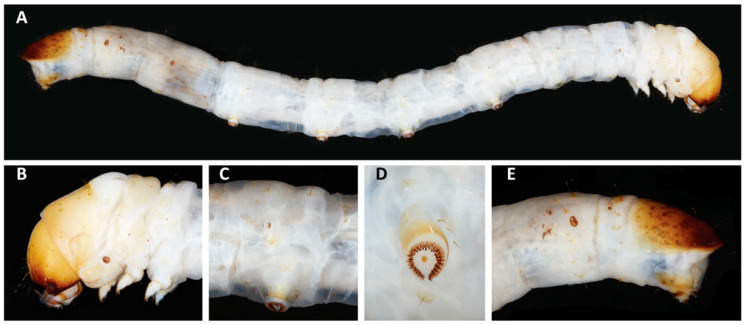
Larva. (**A**) Full side view of Casmara subagronoma larva; (**B**) lateral view of larval head and mouth parts; (**C**) lateral view third abdominal segment; (**D**) ventral view of planta and crochets on sixth abdominal segment; (**E**) lateral view of terminal abdominal segments (Lot USNMENT01328213).

**Figure 7 insects-11-00653-f007:**
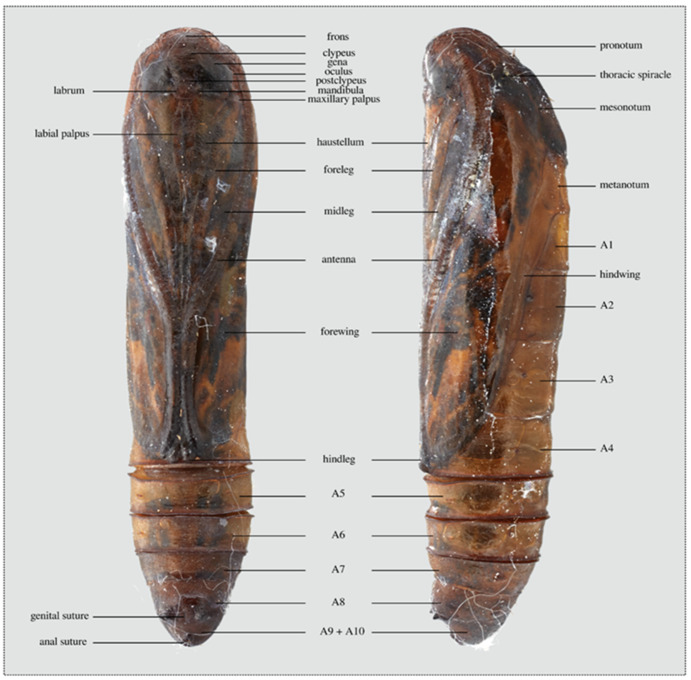
Pupa with labeled parts.

**Figure 8 insects-11-00653-f008:**
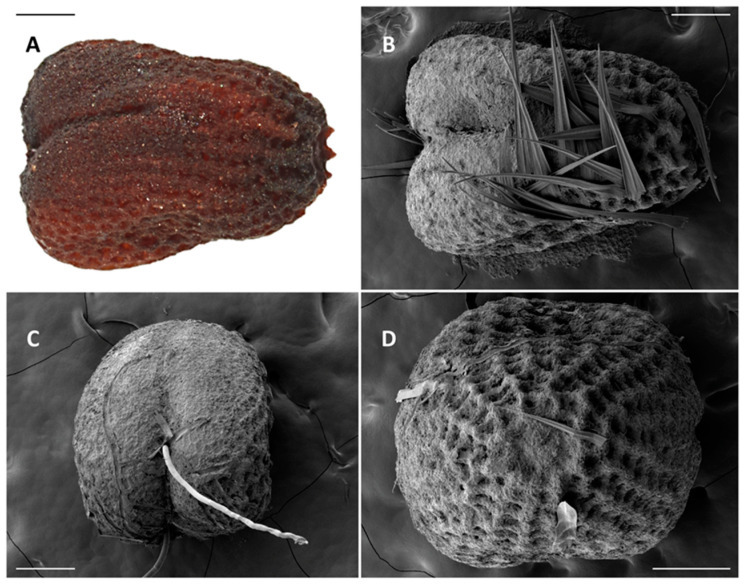
Egg of *C. subagronoma* (**A**) in color; (**B**) longitudinal groove bisecting the wide end; (**C**) longitudinal groove, top view; (**D**) micropile depression surrounded by blunt points. Measure bar = 0.2 mm.

**Figure 9 insects-11-00653-f009:**
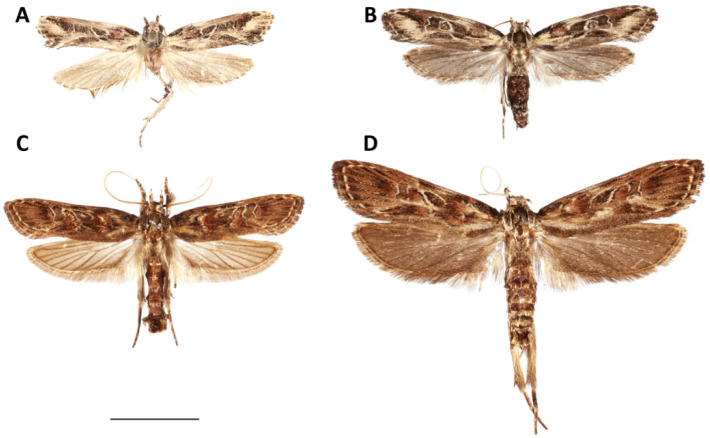
Dorsal view of adult (**A**) *C. subagronoma*, male (USNMENT01328210); (**B**) *C. subagronoma*, female (USNMENT01328218); (**C**) *C. agronoma*, male (USNMENT01200872); (**D**) *C. agronoma*, female (USNMENT01200871). Measure bar = 10.0 mm.

**Figure 10 insects-11-00653-f010:**
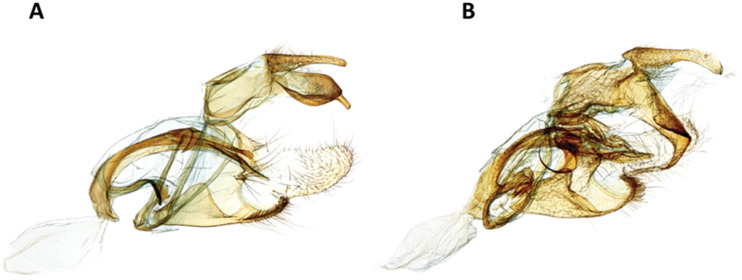
Medial view male genitalia (**A**) *Casmara subagronoma* (USNMENT01328210, USNM slide # 146,475); (**B**) *C. agronoma* (USNMENT01200872, USNM slide # 146,474).

**Figure 11 insects-11-00653-f011:**
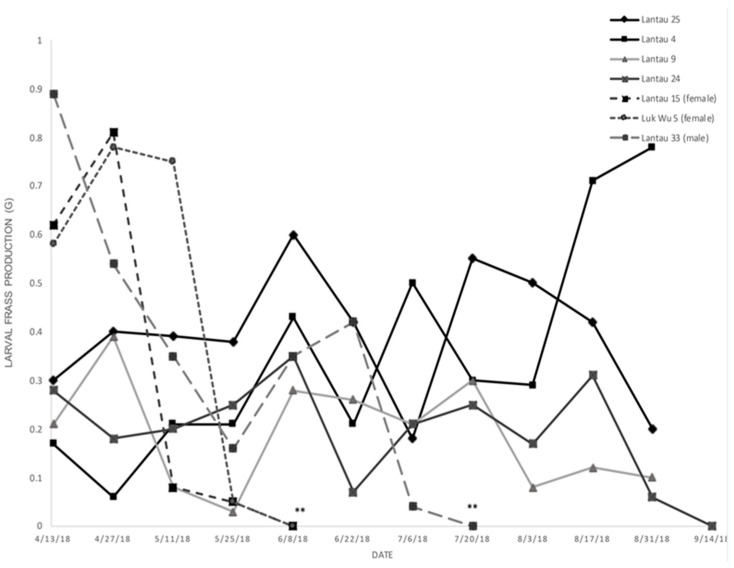
Bi-weekly frass production (dried g) of *C. subagronoma* larvae. Dashed lines are from late-instar larvae (**) that pupated (signaled by quiescence of frass production).

**Table 1 insects-11-00653-t001:** No-choice host range test of *C. subagronoma* on *Myrcianthes fragrans* and *Myrtus communis*. Larvae were transferred to cut stems of non-targets and their feeding activity was tracked until death or emergence.

Host	Larva lShipment Number *	Test Initiated	Larva ID	Date Individual Observed Dead	Stage at Death	Cause of Death	Last Date Individual Observed Alive †	Stage at Observation	Date Pupa	Date Adult	Sex	Maximum Lifespan (Days)
*Rhodomyrtus tomentosa*	2015-1008	19-Jan-16	1	1-Jun-16	larva	parasitoid	22-Apr-16	larva				134
	2015-1008	19-Jan-16	5	17-Feb-16	larva	transfer injury	17-Feb-16	larva				29
	2015-1008	19-Jan-16	16	22-Mar-18	larva	diminished feeding	22-Mar-18	larva				793
	2015-1008	19-Jan-16	31	5-Apr-18	larva	trapped in cage fold	5-Apr-18	larva	unknown	unknown	male	807
	2015-1008	19-Jan-16	38	30-Sep-17	larva	failed to bore after transfer	30-Sep-17	larva				620
*Myrtus communis*	2015-1008	19-Jan-16	29	1-Apr-16	larva	diminished feeding	15-Mar-16	larva				73
	2015-1008	19-Jan-16	44	22-Feb-16	larva	rejected stem	9-Feb-16	larva				34
	2015-1008	19-Jan-16	30	12-Dec-16	larva	diminished feeding	7-Nov-16	larva				328
	2015-1008	19-Jan-16	9	14-Apr-18	larva	bored to soil level and died	10-May-17	larva				816
	2015-1008	19-Jan-16	8	12-Jun-17	larva	diminished feeding	10-May-17	larva				510
*Rhodomyrtus tomentosa*	2016-1001	14-Sep-16	12	8-Dec-16	larva	parasitized	15-Nov-16	larva				85
	2016-1001	14-Sep-16	9	19-Sep-16	larva	unknown	12-Sep-16	larva				5
	2016-1001	14-Sep-16	22	11-Jun-18	larva	unknown	27-Sep-17	larva				635
*Myrtus communis*	2016-1001	14-Sep-16	1	12-Nov-16	larva	parasitized	14-Sep-16	larva				59
	2016-1001	14-Sep-16	3	16-Dec-16	larva	parasitized	27-Oct-16	larva				93
	2016-1001	14-Sep-16	6	27-Sep-17	larva	failed to bore after transfer	25-Jun-17	larva				378
*Rhodomyrtus tomentosa*	2015-1008	17-Feb-16	6	29-Jun-16	larva	parasitoids	6-Jun-16	larva				133
	2015-1008	16-Feb-16	28	18-Oct-16	larva	parasitoids	27-Sep-16	larva				245
	2015-1008	17-Feb-16	33	17-Aug-17	larva	diminished feeding	25-Jul-17	larva				547
*Myrcianthes fragrans*	2015-1008	17-Feb-16	4	26-Jul-16	adult		17-Jul-16	adult	17-Jun-16	17-Jul-16	male	160
	2015-1008	16-Feb-16	25	6-Jun-16	larva	diminished feeding	3-Jun-16	larva				111
	2015-1008	17-Feb-16	43	6-Jun-17	pupa	pupal orientation ‡	14-Apr-17	pupa				475
*Rhodomyrtus tomentosa*	2016-1001	14-Sep-16	8	28-Sep-17	larva	failed to bore after transfer	7-Jun-17	larva				379
	2016-1001	14-Sep-16	18	12-Jun-18	larva	unknown	6-Jun-17	larva				636
	2016-1001	14-Sep-16	10	11-Jun-18	larva	plant died	6-Jun-17	larva				635
*Myrcianthes fragrans*	2016-1001	14-Sep-16	2	27-Sep-17	larva	failed to bore after transfer	26-Sep-17	larva				378
	2016-1001	14-Sep-16	19	5-Aug-17	adult			adult	13-Jul-17	5-Aug-17	male	325
	2016-1001	14-Sep-16	15	8-Jun-17	adult			adult	11-May-17	8-Jun-17	male	267

* Larvae tested from Shipment 2015-1008 were reared on R. tomentosa in the laboratory from 1 Dec 2015 to start of test on 17 Feb 2016. * Larvae tested from Shipment 2016-1001 were reared on R. tomentosa in the laboratory from 9 Mar 2016 to the start of test on 14 Sep 2016. † Observation dates vary because larvae/pupae are observed only when each individual’s stem needs to be changed. ‡ Pupal orientation appeared to be reversed and may have caused death.

**Table 2 insects-11-00653-t002:** Comparison of proximal stem diameter and stem bore diameter of 2018 field-collected *C. subagronoma* larval stems. The larvae were transferred to Florida *R. tomentosa* and *C. sinensis* on March 15 (*N* = 3/species), March 16 (*N* = 5/species), March 19 (*N* = 1 to *C. sinensis*), and March 23 (*N* = 1 to *R. tomentosa*) and their subsequent frass production and longevity under greenhouse conditions.

Hong Kong Collection Sites 12 March 2018	Larva ID	Measurements of Hong Kong Damaged Stems Containing Larvae	Host Plant	Dried Frass Weight (g) After Transfer to Florida Host Material	Larval Lifespan in Laboratory (Weeks)
Stem Diam (mm)	Larval Bore Diam (mm)	15 March to 13 April	14 April to 27 April
Lantau	2	3.75	1.13	*R. tomentosa*	0.03	0.03	39
	4	2.98	1.37		0.14	0.03	>24
	9	3.79	1.51		0.24	0.36	>24
	12	3.86	1.84		0.27	0.18	23
	15	4.6	2.07		0.57	0.78	30
	20	3.19	1.55		0.33	0.22	39
	24	3.96	1.58		0.25	0.14	>24
	25	3.96	1.27		0.30	0.39	>24
	10	3.68	1.68		0.38	0.29	39
	mean	3.75	1.56		0.28	0.27	
	stdev	0.47	0.29		0.15	0.23	
	SE	0.16	0.10		0.05	0.08	
Lantau	1	4.43	1.67	*C. sinensis*	0	0	<4 weeks
	3	4.43	1.59		0.009	0	<6 weeks
	6	3.32	1.68		0.002	0	<4 weeks
	11	3.62	1.63		0.001	0	<4 weeks
	14	3	1.63		0.001	0	<4 weeks
	16	3.06	1.32		0.002	0	<4 weeks
	22	3.92	1.37		0	0	<4 weeks
	27	3.79	1.15		0.002	0	<4 weeks
Luk Wu	1	3.86	2.12		0.002	0	<4 weeks
	mean	3.71	1.57		0.002	0	
	std	0.52	0.28		0.003	0	
	SE	0.19	0.10		0.001	0

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
