# Peer review of "The Biology of Casmara subagronoma (Lepidoptera: Oecophoridae), a Stem-Boring Moth of Rhodomyrtus tomentosa (Myrtaceae): Descriptions of the Previously Unknown Adult Female and Immature Stages, and Its Potential as a Biological Control Candidate"

_insects, 2020, doi:10.3390/insects11100653_

Round 1
Reviewer 1 Report
Unlike most manuscripts in the journal, this was more descriptive than "hypothesis testing". Nevertheless, it is well written, important, and very detailed. I have no comments or concerns regarding publishing this in the journal. My comments below are very minor and can be addressed easily.
Line 79: Rephrase the sentence. Not clear how "Walker" fits into to the sentence. Table 1 :Text too small; cant read
Figure 6: same concerns
Is it possible to show a picture of the host plant, and other hosts that were tested on?
Author Response
Thank you for your thoughtful suggestions.
Line 79: "Walker" is in reference to the genus epithet and is used only in this first instance.
Table 1 has been reformatted as landscape to accommodate a larger font.
Figure 6 has been increased in size to accommodate larger labels on the figure.
An additional figure has been added to manuscript (as Figure 1) to illustrate an invaded area of Rhodomyrtus tomentosa in Florida and a closeup of the flower. Thank you for the suggestion.
Reviewer 2 Report
Review of Journal: Insects
Manuscript ID: insects-929838
Type of manuscript: Article
Title: The Biology of Casmara subagronoma (Lepidoptera: Oecophoridae), a Stem
Boring Moth of Rhodomyrtus tomentosa (Myrtaceae): Descriptions of the
Previously Unknown Adult Female and Immature Stages, and its Potential as a
Biological Control Candidate
Authors: Susan A. Wineriter-Wright, Melissa C. Smith *, Mark A. Metz, Jeffrey
- Makinson, Bradley T. Brown, Matthew A. Purcell, Kane L. Barr, Paul D.
Pratt
Overview.
This is a substantial contribution to systematic entomology and basic insect-plant literature pertinent to biological control. The introduction is well written, informative, and detailed about the biology of this insect and its congeners. The introduction moves on to the relationships between plant hosts, species of tea and the stem borers in the genus Casmara in their broad native range in Asia. The detail about plant damage is unusual and useful to the general study of insects on plants, as is that about the life history of the insect in the context of the host plant. Information on host range is equally detailed.
The materials and methods are well explained and complete, in my estimation. The images are clear and instructive.
The Results are well written and given in substantial detail. Figures are all useful and well done.
The Discussion ties the threads of the present study together with extensions to the difficulty of rearing and challenges to use for biological control of this insect, were it not of danger to native shrubs of the new world. The review of stem borers in biological control is informative and useful.
Author Response
Thank you for your thoughtful comments. We very much appreciate it.
Reviewer 3 Report
This manuscript by Wineriter-Wright et al. deals with the feeding biology and provides a detailed morphological description of Casmara subagronoma. Even though their results demonstrated this species to be unsuitable as a biological control agent this study still has considerable merit. Studies of this scope and depth are very rare for oecophorid moths which makes this manuscript a valuable source of data for many fields of research beyond the control of invasive species.
The manuscript is very polished, reporting of results and their discussion is concise while still being comprehensive. I have no specific comments, this manuscript can be published in its present state.
Author Response
Thank you for your thoughtful comments. We very much appreciate your feedback about our manuscript.